# Prebiotics Progress Shifts in the Intestinal Microbiome That Benefits Patients with Type 2 Diabetes Mellitus

**DOI:** 10.3390/biom13091307

**Published:** 2023-08-25

**Authors:** Luis Vitetta, Nick N. Gorgani, Gemma Vitetta, Jeremy D. Henson

**Affiliations:** 1Faculty of Medicine and Health, The University of Sydney, Sydney, NSW 2006, Australia; 2OzStar Therapeutics Pty Ltd., Pennant Hills, NSW 2120, Australia; 3Gold Coast University Hospital, Southport, QLD 4215, Australia; 4Prince of Wales Clinical School, University of New South Wales, Sydney, NSW 2052, Australia

**Keywords:** prebiotics, intestinal microbiome, short chain fatty acids, butyrate, functional foods, Type 2 Diabetes Mellitus

## Abstract

Hypoglycemic medications that could be co-administered with prebiotics and functional foods can potentially reduce the burden of metabolic diseases such as Type 2 Diabetes Mellitus (T2DM). The efficacy of drugs such as metformin and sulfonylureas can be enhanced by the activity of the intestinal microbiome elaborated metabolites. Functional foods such as prebiotics (e.g., oligofructose) and dietary fibers can treat a dysbiotic gut microbiome by enhancing the diversity of microbial niches in the gut. These beneficial shifts in intestinal microbiome profiles include an increased abundance of bacteria such as *Faecalibacterium prauznitzii*, *Akkermancia muciniphila*, *Roseburia species*, and *Bifidobacterium* species. An important net effect is an increase in the levels of luminal SCFAs (e.g., butyrate) that provide energy carbon sources for the intestinal microbiome in cross-feeding activities, with concomitant improvement in intestinal dysbiosis with attenuation of inflammatory sequalae and improved intestinal gut barrier integrity, which alleviates the morbidity of T2DM. Oligosaccharides administered adjunctively with pharmacotherapy to ameliorate T2DM represent current plausible treatment modalities.

## 1. Introduction

The intestinal microbiota presents a complex ecosystem consisting of bacteria, enteric viruses, archaea, fungi, and protozoa [1,2]. The structure of the intestinal microbiota is influenced by host genetics and environmental factors. Recent studies have shown that the gut microbiota exhibits a complex contribution to overall health, by actively regulating metabolic functions through genes, proteins, and metabolites. Furthermore, multiple factors contribute to the establishment of the human gut microbiota during infancy [3]. The main factors that are the predominant drivers in shaping the intestinal microbiome include diet across a lifetime; intestinal bacteria have a pivotal role in maintaining a metabolic and immune function equilibrium that protects against pathogenic insults.

Any subsequent alterations in the composition of the gut bacterial profile (dysbiosis) have been associated with the pathogenesis of numerous inflammatory diseases and infections [3]. Moreover, based on current evidence, it has been recently postulated and encouraged that novel strategies should integrate the administration of probiotics with conventional anti-cancer therapies [4].

The current scientific literature suggests that as the intestinal microbiota composition is observed to differ between individuals and is contingent on a variety of factors such as genetics and diet, a percentage of individuals may harbour intestinal bacteria that have been reported to be associated with pro-inflammatory effects, while others may harbour intestinal bacteria with anti-inflammatory effects [5]. Biochemical techniques that allow for the enhanced characterization of the intestinal microbiome have provided further evidence of the pro-inflammatory and anti-inflammatory nature of the gut cohort of bacteria in health and disease [5]. The continuous improvement in knowledge relevant to the inflammatory pathways that interact with bacteria will further elucidate factors that underpin the varying presentations of the same disease and the varied responses reported to the same treatment in different individuals [5]. What becomes biologically and clinically plausible is the administration of anti-inflammatory microbes that can be formulated in probiotic formulations and used in therapies with or without prebiotics [5].

The administration of oral probiotic bacteria has been demonstrated to interact with numerous intestinal structures, including intestinal epithelial cells and immune cells associated with the lamina propria, through Toll-like receptors that then induce the production of a plethora of cytokines/chemokines [6,7,8]. Macrophage chemoattractant protein 1, produced by intestinal epithelial cells, produces signals that are interpreted by other immune cells, leading to the activation of the mucosal immune system. Such activation has been reported to be characterized by increased immunoglobulin A+ cells of the intestine, bronchus, and mammary glands and the activation of T cells [6,8].

Furthermore, probiotics have been reported to activate regulatory T cells that release an anti-inflammatory interleukin (i.e., IL-10). What has been reported is that probiotics have multiple effects that reinforce the intestinal epithelial barrier integrity, coupled with increases in mucins, tight junction proteins, and Goblet and Paneth cells [6,7]. Additionally, probiotics have been proposed to modulate the intestinal microbiota by maintaining a healthy pro-inflammatory to anti-inflammatory balance that continuously suppresses the overgrowth of potential pathogenic bacteria in the intestines. Alternatively, it has been demonstrated that the long-term administration of probiotics does not affect the intestinal homeostasis. The viability of probiotics is crucial in their interactions with intestinal epithelial cells and macrophages, which are skewed toward a favorable innate immune response. The important role of macrophages in the immune response progresses without inducing an inflammatory pattern, with minor increases in the cellularity of the lamina propria [6,7].

Of interest is the role that probiotics may have in activating the microbicidal actions of peritoneal and spleen macrophages to protect against the action of different pathogens [6]. Furthermore, it has been also reported that in malnutrition models, such as undernourishment and obesity, probiotic formulations have been able to increase both intestinal and systemic immune responses. Additionally, probiotic bacteria may also contribute to recovering the histology of both the intestine and the thymus that have been damaged due to poor nutrition choices. The administration of probiotic bacteria has a long history, and the emerging evidence shows that it is a safe and natural strategy for allergy prevention and treatments [6]. Moreover, different mechanisms such as those proposed to generate cytokines from activated pro-T-helper type 1 cells have been reported to favor the production of IgG instead of IgE [6].

A recent review reports that intestinal microbiota dysbiosis has been shown to have adverse health effects that will lead to a variety of chronic disease progressions [9]. The subsequent regulation and mechanism of involvement of the intestinal microbiota have been reported in neurodegenerative diseases, cardiovascular diseases, metabolic diseases, and gastrointestinal diseases [9]. Furthermore, a recent review has reported that experimental models of inflammatory bowel disease have strongly suggested that while intestinal bacteria often progress immune activation, chronic inflammatory responses in turn can shape the intestinal microbiota and contribute to dysbiosis in inflammatory bowel diseases [10].

It is generally accepted that in genetically susceptible individuals, inflammatory bowel diseases can result from the dysregulation of immune responses to environmental and/or microbial agents [11]. Of interest is a recent laboratory murine study that administered the antibiotics rifaximin (i.e., a dose of 50 mg/kg/dose) and/or Mutaflor (10^9^ CFU/dose) given intragastrically once a day to investigate the healing effect of acetic acid-induced colitis [12]. The study reported that the antibiotic significantly accelerated the healing of colonic damage. Moreover, the administration of the antibiotic significantly reduced the concentration of the proinflammatory cytokine TNF-α, as well as the activity of myeloperoxidase in the colonic mucosa of rats [12]. Mutaflor (i.e., a probiotic formulation) given alone had no significant effect on the activity of colitis. However, when Mutaflor was administered in combination with the antibiotic, there was a significantly enhanced therapeutic effect of the antibiotic. The authors concluded that the antibiotic and probiotic formulations exhibited synergic anti-inflammatory and therapeutic effects in acetic-acid-induced colitis in rats [12].

A growing body of evidence reports that intestinal bacteria can influence a plethora of hormones [13]. For example, decreased abundance in microbial diversity has been linked with high thyroid stimulating hormone (TSH) levels [14]. Furthermore, an increase in TSH concentrations may cause the production of lower levels of thyroid hormones T3 and T4, which can at times progress to hypothyroidism. An imbalanced intestinal microbiome has also been associated with hypothyroidism [15]. Studies document that common symptoms of hypothyroidism include weight gain, sensitivity to cold, dry skin, constipation, and poor memory [16]. The hormone estrogen is also subject to bacterial actions. Studies report that the intestinal microbiota is a key regulator of the level of circulating estrogen in the systemic circulation [17]. In addition, the intestinal microbes can produce beta-glucuronidase, an enzyme that can convert estrogen into its active forms [17]; it was also noted that intestinal dysbiosis can alter the amount of active estrogen in the circulation. Furthermore, research reports that there exists a specific group of bacteria termed the *estrobolome* [18]. The *estrobolome* has been reported to consists of bacterial genes that are capable of metabolizing estrogens, an important factor given that estrogen is a potent promoter of tissue growth throughout the body [18].

Another important metabolite that is associated with gut bacteria is serotonin [19]. Serotonin is a precursor for the formation of melatonin, a hormone that has been reported to regulate mood [19]. Of significant clinical interest is that studies have reported that shift workers tend to present with gut dysbiosis. Hence, insufficient sleep can cause negative effects on the intestinal microbiome [19]. Further, shift work has been shown to increase inflammatory responses and the risk of obesity, metabolic syndrome, and T2DM.

Additional hormones such as cortisol, epinephrine, and norepinephrine are hormones that are associated with an alert response [20]. These hormones are released during the flight-or-flight response, leading to heart rate and blood pressure elevations, which can subsequently increase hormone levels. However, when these hormone levels remain high for prolonged periods, the dynamics of the intestinal microbiome can be altered [20]. High levels of stress hormones have also been shown to trigger adverse gene expression in gut microbes [20]. In animal studies, high epinephrine and norepinephrine levels can stimulate infections by increasing the virulence of some pathogens [20].

A chronic increase in intestinal permeability that has been termed leaky gut has been observed in the patients and animal models of metabolic diseases (e.g., T2DM) [21]. This state often correlates with a metabolic disease state [22]. Recent reports have documented that the intestinal microbiota affects intestinal and systemic health via bacterial-generated metabolites, especially short-chain fatty acids and lipopolysaccharides, which can trigger and maintain a leaky gut [22].

The endotoxin lipopolysaccharide (LPS) is a component of Gram-negative bacteria and is responsible for sepsis and neonatal mortality [23], yet in experimental studies, low levels of LPS show tissue protection. In a recent murine study, the effects of LPS, which was applied to suckling rats on the pancreas of adult animals, were investigated [23]. The study concluded that the endotoxemia that was induced in suckling rats by the intraperitoneal application of LPS from *Escherichia coli* or *Salmonella typhi* in the early period of life with LPS reduced histological manifestations of acute pancreatitis. Moreover, pancreatic weight and plasma lipase activity were decreased, and SOD concentration was reversed and accompanied by a significant reduction in lipid peroxidation products in the pancreatic tissue [23]. In the pancreatic acini, there were observed significant increases in protein signals for toll-like receptor 4 and for heat shock protein 60. In addition, the signal for the CCK1 receptor was reduced and pancreatic secretory responses to caerulein were decreased, whereas basal enzyme secretion was unaffected [23]. Hence exposing suckling rats to endotoxin had an impact on the pancreas in the adult organism.

T2DM is a group of chronic endocrine and metabolic disorders characterized by defects in insulin production, secretion, and signaling that are insufficient to maintain a balanced blood glucose level. The interaction between the gut microbiome, dietary practices, and the activity of mucosal immunity influences the progression of T2DM. A significant body of research evidence has documented that the intestinal microbiome is associated with metabolic disease and that the progression of intestinal dysbiosis leads to gut metabolic dysregulation [24,25].

Studies have shown that functional foods administered as prebiotics and/or probiotics have led to beneficial alterations of the intestinal microbiome in T2DM, hence improving glycemic control [26,27]. In addition, recent studies explored the co-administration of probiotics and prebiotics as adjuncts to pharmacotherapy, also providing an effective treatment strategy in managing hyperglycemia symptoms in T2DM [27,28].

Functional foods from plants have a long history of promoting health benefits, primarily due to the supply of compounds that, at least in part, can offer protective effects, as exemplified by fruit and vegetable-rich diets [29]. The study of functional foods in human nutrition has become a fundamental issue in the prevention and management of metabolic disorders such as T2DM.

## 2. Pathogenesis of Diabetes Mellitus: Link between the Intestinal Microbiome Dysbiosis and Insulin Resistance

The exact factors that drive the development of metabolic diseases remain elusive, as most metabolic diseases can be associated with both a Western-type diet and adverse microbiome modifications such as dysbiotic gut. These intestinal microbiome shifts are characterized by reduced diversity, enrichment of opportunistic pathogens, and an imbalanced ratio of beneficial to detrimental metabolites [30]. Various distinct pathophysiologic abnormalities have been associated with T2DM. A decreased peripheral glucose uptake (i.e., mainly from skeletal muscle) in combination with an augmented endogenous glucose production are the characteristic hallmark features of insulin resistance [31,32]. Furthermore, increased lipolysis, elevated systemic free fatty acid levels, and the accumulation of intermediary lipid metabolites strongly contribute to further increases in glucose output, reducing peripheral glucose utilization, and impairment of beta-cell function [32,33]. A recent review summarized data from epidemiological studies that noted that patients diagnosed with T2DM have a two-fold increased risk for non-alcoholic fatty liver disease (NAFLD) development and vice versa [31]. The presence of NAFLD] is now considered an integral part of the insulin resistant state [31]. The traditional concepts of *glucotoxicity* and *lipotoxicity*, which encompass the process of beta cell deterioration in response to chronic elevations in glucose and lipids, also include all nutrients, giving rise to the term *nutri-toxicity* [32]. The delayed transport of insulin across the microvascular system is also partially responsible for the development of tissue insulin resistance.

Furthermore, Solis-Herrera and colleagues [32] report that compensatory insulin secretion by pancreatic beta cells could initially maintain normal plasma glucose levels. Yet they also report that beta cell function is already compromised and abnormal during the early stages of T2DM and progressively becomes aggravated over time.

Solis-Herrera and colleagues have suggested that during the post-prandial period, there is an inappropriate release of glucagon from the alpha cells of the pancreas [32]. Consequently, Solis-Herrera et al. posit that this is probably due to impaired insulin secretion as well as an excessive secretion of glucagon in patients diagnosed with T2DM. These biochemical effects would be secondary events to an incretin defect, where the latter is defined as a primary inadequate release or response to the gastrointestinal incretin hormones upon meal ingestion. Therefore, it is interesting to note that the intestinal microbiome may be complicit in the hormonal and metabolic disturbances that ensue and that are observed in T2DM.

Glucose toxicity research refers to higher than normal concentrations of glucose in cells and tissues that have been associated with adverse health effects [32,34]. The most common effect reported is that glucose toxicity presents as the ability of excess circulating systemic glucose to impair insulin secretion and action. A resultant aggravated cycle of declined glycemic control in T2DM ensues [32,34]. Glucose toxicity has been expanded to encompass other effects of hyperglycemia such as the complications of diabetes [32]. Furthermore, it has been suggested that such terminology (i.e., glucose toxicity) be applied to those adverse effects of high glucose concentrations that are specific for glucose or its metabolites (i.e., are not reproduced by hyperosmolality) [34]. McClain has also noted that the term toxicity may be misleading as some of the ultimately harmful effects of excess systemic glucose levels may represent normal physiologic responses rather than a significant deleterious biochemical effect on cells by non-physiologic mechanisms [34].

Moreover, investigations relevant to the intestinal organoids have been used to analyze the differentiation of enteroendocrine cells and to manipulate their density for treating T2DM [35]. Such molecular mechanisms may have direct clinical implications. In addition, Filipello and colleagues [35] recently showed that glucotoxicity can impair L-cell differentiation, an important factor associated with decreased intestinal stem cell proliferative capacity [35]. We note that this study provided the identification of new targets involved in new molecular signaling mechanisms that may be impaired by glucotoxicity and that these targets could be a useful tool for the treatment of T2DM [35]. Obesity is a risk factor for various metabolic disorders such as T2DM and cardiovascular disease [36]. The two dominant phyla in the intestines comprise the *Firmicutes* and *Bacteriodetes* [37]. Reports show an existing link between the intestinal microbiome and obesity, with an attendant increase in the abundance of *Firmicutes* and a decrease in the diversity of *Bacteroidetes* [37,38]. Obesity and the complicit intestinal microbiome in T2DM links host molecules that induce intestinal microbiota dysbiosis and immune and pro-inflammatory responses, with intestinal microbial metabolites being involved in the pathogenesis of the disease [39]. The T2DM metabolic syndrome of abnormal lipid and glucose metabolism is associated with numerous dysfunctional physiological complications [40]. T2DM has been characterized by pancreatic beta-cell dysfunction and peripheral insulin resistance, leading to defects in glucose metabolism [41]. In addition, a dysbiotic intestinal microbiome has been reported to be complicit in the pathogenesis of T2DM by maintaining the gut in a chronic low-grade inflammatory state [39,42]. The earliest proposed strategy to manipulate the intestinal microbiome for a health benefit was the use of probiotics [43]. A recent review has directed new perspectives in the treatment of T2DM, with the goal being to achieve an effective adjustment of blood glucose and blood lipid levels as well as body weight in patients diagnosed with T2DM to healthy profiles [44]. The administration of synbiotic formulations (i.e., prebiotics + probiotics) can also improve glucose and lipid profiles in patients diagnosed with T2DM [44]. Studies with prebiotics (without probiotics) have also shown efficacy in patients diagnosed with T2DM (Table 1).

Could prebiotics that significantly influence the gut microbiome, administered as adjuvants to pharmacotherapy, maintain favorable indexes of lipid profile, blood pressure, and fasting blood glucose in patients with T2DM? The application of adjuvant prebiotics with oral hypoglycemic medications could be a new management strategy for lipid and glucose profiles and blood pressure management in T2DM. Two studies with prebiotics and sulfonylureas or metformin have reported improvements in the associated clinical parameters (Table 2). This has become a plausible research initiative given that metformin has been shown in a number of studies with obese and/or T2DM-diagnosed patients to progress gut microbiome shifts favorable to improved glycemic control [45]. Specifically, it was reported that the overall results showed that metformin significantly shifted the proportion of the phyla *Bacteroidetes* and *Verrucomicrobia* and the genera *Akkermansia*, *Bacteroides,* and *Escherichia.* Shifts in the intestinal microbiome remain consistent with the view that gut bacteria influence the hypoglycemic effect of metformin. Mechanistically through the improved integrity of the intestinal epithelial barrier, the production of SCFAs (e.g., butyrate), the regulation of bile acid metabolism, and improved glucose homeostasis [45].

Oligosaccharides are a type of carbohydrate naturally found in many types of plant foods. Oligosaccharides are low-level polysaccharides formed by the linkage of 2–10 monosaccharides through glycosidic bonds. Common oligosaccharides include chitosan oligosaccharides, xylo-oligosaccharides, konjac glucomannan, and brown algae oligosaccharides [46]. In a clinical trial with women diagnosed with T2DM, an oligofructose was reported to improve inflammatory markers [47]. In a systematic review, it was posited that oligosaccharides could offer benefit as a gut microbiome therapeutic strategy for T2DM [48]. In view of this, a recent systematic review and meta-analysis [49] concluded that oligosaccharides can provide beneficial effects for the markers of glycemic control, specifically fasting blood glucose, fasting blood insulin, glycated hemoglobin, homeostasis model assessment of insulin resistance, and quantitative insulin sensitivity index.

### Fiber, Prebiotics, and Intestinal Bacteria

There has been a long appreciation of dietary fiber as being beneficial for intestinal health and improving symptoms of gut progressed inflammatory sequelae [50]. Furthermore, dietary fiber is reported to be a group of heterogenous compounds that are neither digested nor absorbed in the intestines [51]. Categories of fiber, including inulin, fructo-oligosaccharides, and other oligosaccharides, have been included in food labels. Consequently, oligosaccharides have been considered to be the best-known example of a *prebiotic*, being a selectively fermented ingredient that allows specific changes, both in the composition and/or activity of the intestinal microbiome, which may confer health benefits on the host [50].

A recent review tabulates resistant oligosaccharides (e.g., fructans (fructooligosaccharides, oligofructose (OF), and inulin) and galactans) as well documented prebiotics [51]. Additional fibers have been posited as candidates of prebiotics or as having the potential to be classified as prebiotics, while other fibers have no prebiotic effect [51]. Furthermore, studies that have provided evidence for inulin, OF, lactulose, and resistant starch (RS) meet the definition of fibers that encourage bacteria from the *Bifidobacterium* genus. Additionally, isolated carbohydrates and carbohydrate-containing foods, including galactooligosaccharides (GOS), transgalactooligosaccharides (TOS), polydextrose, wheat dextrin, acacia gum, psyllium, banana, whole grain wheat, and whole grain corn, have also been documented as providing a prebiotic effect in the gut [50].

Prebiotics are defined as a group of compounds that have been associated with gut health and that are mediated, at least in part, through degradation products by microorganisms [52]. The chemical structure of prebiotics and the bacterial composition of the intestines determine the fermentation products produced [53,54]. Dietary fiber is found mainly in fruits, vegetables, whole grains, and legumes, and their effects on health mimic those of prebiotics on human health [54]. Numerous compounds have been investigated to determine if, as prebiotics, these functional non-digestible foods can influence the diversity of the intestinal microbiome to promote metabolic health in the gut [53,55] (Table 1). The most common forms of prebiotics that have been studied include fructo-oligosaccharides, galacto-oligosaccharides, and trans-galacto-oligosaccharides [56,57]. These prebiotics are not digested by the host enzymes/tissues. When taken orally, prebiotics travel to the intestines, where they are then metabolized by microorganisms to be used as energy sources [58].

The biochemical process of the fermentation of fiber-rich compounds by the intestinal microbiome yields short-chain fatty acids (SCFAs), including acetate, lactate, butyrate, and propionate [59,60,61]. The net effects of the increased production of SCFAs in the intestines are the lowering of the luminal pH [62], improved availability of calcium and magnesium, and inhibition of potentially pathogenic bacteria [63]. Both *Bifidobacterium* and *Lactobacillus* produce acetate (and lactate), and, as such, contribute to the SCFA-mediated health benefits that prebiotics have been reported to provide. Notwithstanding, members of these genera of microorganisms do not produce butyrate and/or propionate [64]. Moreover, the important function of the bacterial species from the *Bifidobacterium* genus is that these microbes contribute to gut homeostasis and host health. This is achieved through the production of acetate and lactate during carbohydrate fermentation, which are organic acids that in turn can be converted into butyrate by other colon bacteria through cross-feeding interactions [65].

A recent study has investigated how SCFAs modulate the intestinal microbiome and progress their functionality [66]. The results provided important information on how prebiotics elevate the colonic level of specific SCFAs for therapeutic benefit. Various human studies with healthy children and adults have demonstrated the effect that prebiotics have on the abundance and dynamics of microbial species in the intestines (Table 1). Clinical studies that have been conducted have also reported significant favorable shifts in the intestinal microbiome. In most of the studies, shifts towards increases in the abundance of *Bifidobacterium* and *Lactobacillus* genera have been reported.

**Table 1 biomolecules-13-01307-t001:** Prebiotics shift the intestinal microbiome: human studies.

PREBIOTICS (n) REFERENCE	TREATMENT TIME	DOSE ADMINISTERED	FLUID	FAECAL INTESTINAL MICROBIAL SHIFTS BY PREBIOTICS VERSUS CONTROLS
**STUDIES WITH CHILDREN**				
PDX/FOS * [n = 77] [67]	2-weeks	4.17 g PDX| 0.45 g FOS O.D.	in water	↑ *Bifidobacterium* and *Lactobacillus*
AX [n = 10] [68]	48 h	10 g/L fluid	in water	↑ *Lactobacillus*
**STUDIES WITH ADULTS**				
Inulin [n = 17 W] [69]	8 days	20 g/day to 40 g/day	in water	↑ *Bifidobacterium* ↓ *Enterococci*|*Enterobacteriaceae*
Inulin [n = 10] [70]	2-weeks	8 g/day B.I.D.	in water	↑ *Bifidobacterium*|↑ *Clostridia*
GOS [n = 18] [71]	3-weeks	2.5 g/day 5 g/day 10 g/day	edible chews	↑ abundance *Bifidobacterium* ↑ *Faecalibacterium prausnitzii* ↓ *Bacteroides*
PDX [n = 20 M] [72]	3-weeks	21 g/day O.D.	mixed food	↑ *Faecalibacterium* | *Phascolarctobacterium*| ↑ *Dialister*
PDX [n = 15] [73]	3-weeks	8 g/day O.D.	powder	↑ *Ruminococcus intestinalis*| *Clostridium clusters* I, II and IV, ↓ abundance *Lactobacillus*| ↓ *Enterococcus* group
Lactulose [n = 12] [74]	4-weeks	20 g/day B.I.D.	mixed food	↑ *Bifidobacterium*|*Lactobacillus*
Synthetic 2′-O-fucosyllactose and/or lacto-N-neotetraose [n = 44] [75]	2-weeks	5 10 20 g/day	Dietary bars	↑ *Actinobacteria* | *Bifidobacterium* ↓ *Firmicutes*|*Proteobacteria*
Wheat Bran or Maltodextrin [n = 20] [76]	3-weeks Cross-over study	10 g/day	Dietary bars	↑ *Bifidobacterium adolescentis*
PDX + SCF or placebo [n = 21] [77]	3-weeks Cross-over study	21 g/day PDX 21 g/day SCF	Dietary bars	With increased total corn fiber consumed from 10 g to 50 g/day Optimum mean daily consumption of 30–35 g/day Shift of the *Bacteroidetes*:*Firmicutes* ratio positively correlated to total dietary fiber intake.
Soluble Corn Fiber [n = 28 F] [78]	4-weeks/dose [12-weeks]	0 10 20 g/day	in water	↑ *Clostridium* ↑ unclassified *Clostridiaceae*
XOS [n = 12 W [n = 11 M]] [79]	8-weeks	1.4 g/day or 2.8 g/day O.D.	capsule	↑ *Bacteroides fragilis*|*Bifidobacterium*
XOS [n = 13] [80]	3-weeks	4 g/day	mixed food	↑ *Bifidobacterium* species
Inulin or Maltodextrin [n = 44] [81]	8-weeks Cross-over study	12 g/day	in water	↑ relative abundances *Anaerostipes*|Bilophila|*Bifidobacterium*
Crystalline Maize [Treat Arm 1} [n = 5 W 5 M]	4-weeks dose escalation	0 10 20 35 50 g/day 10 g/d to 50 g/d	powder sachets in water	Individualized effects with ↑ *B. adolescentis*|*Parabacteroides* spp.| ↑ *Eisenbergiella* spp. (Maize)
Tapioca RS4 [n = 5 W 5M] [Treat Arm 2] Digestible Corn Starch [n = 5 W 5 M] [Placebo Arm] Potato RS4 [n = 5 W 5 M] [82] [Treat Arm 3]	4-weeks dose escalation 4-weeks dose escalation	0 10 20 35 50 g/day 10 g/d to 50 g/d 0 10 20 35 50 g/day 10 g/d to 50 g/d	powder sachets in water powder sachets in water	↑ *E. rectale* (Maize)↑ *P. distasonis*|*Clostridiales * (Tapioca extracted from cassava root) — → *E. rectale * → *P. distasonis*

* O.D. = once per day; PDX/FOS = polydextrose/fructooligosaccharide; B.I.D. = twice per day; F = females; M = males; XOS = xylooligosaccharide; SCO = soluble corn fiber; RS4 = type-IV resistant starches; ↑ = increased; ↓ = decreased; → = no effect.

Other prebiotic compounds from dietary fiber can also encourage the production of butyrate and/or propionate in the intestines [83], which has been studied for its effects on human health. Consequently, prebiotics have been shown to increase significantly the colonic luminal concentration of SCFAs such as butyrate [54]. Butyrate is the primary energy source of differentiated intestinal epithelial cells (Figure 1), as its metabolism serves to consume oxygen, permitting the preservation of an anaerobic gut luminal environment that maintains the fitness of the strict anaerobes populating that part of the intestines [84].

In order to maintain homeostasis, the colonic epithelia are skewed toward oxidative phosphorylation, which results in high epithelial oxygen consumption [84]. Consequently, the resultant intestinal colonic epithelial hypoxia supports the maintenance of a local microbial community that is obligate anaerobic bacteria dominant, providing a beneficial conversion of fiber into fermentation products that can be absorbed by the host [84]. Alternatively, those conditions that alter the metabolism of the colonic epithelium, which increases epithelial oxygenation that drives an expansion of facultative anaerobic bacteria, can lead to gut dysbiosis in the colon. Enteric pathogens can then subvert colonocyte metabolism to escape the niche protection conferred by the gut microbiota [84].

Butyrate has also been reported to act as a histone deacetylase (HDAC) inhibitor, exhibiting pleiotropic modifications in biochemical transcription and host physiology [87] and exerting an influence on the activation of anti-inflammatory Treg cells [84].

Consequently, SCFAs have been reported to have multiple effects on the host. For example, reports have documented that butyrate can stimulate the synthesis of host defense peptides via intestinal epithelial cells [88]. The mechanism is via the interactions of butyrate with G protein-coupled receptor 43 (GPR43). GPR43 activates the Jun N-terminal kinase, mitogen-activated protein kinase, extracellular signal-regulated kinase pathways, and cell growth pathways [89]. Butyrate can also influence intestinal epithelial development and barrier integrity [59]. Given that SCFAs can diffuse into the systemic circulation through gut enterocytes, this translocation allows metabolites generated from the bacterial fermentation of prebiotics to affect not only the intestinal physiology but also that of distant site organs (e.g., the liver) [18].

Other SCFAs such as propionate can affect a subset of CD4^+^ T helper cells: the T helper 2 (T_H_2) cells [90]. T_H_2 cells facilitate the activation and maintenance of antibody-mediated immune responses against extracellular parasites, bacteria, allergens, and toxins [90]. T_H_2 cells mediate these functions by co-operating and/or producing various cytokines such as IL-4, IL-5, IL-6, IL-9, IL-13, and IL-17E (IL-25) [90]. This, in turn, may stimulate macrophages in the respiratory tract as well as dendritic cells in the bone marrow [91]. Peptidoglycan (i.e., the basic structure in nearly all bacterial cell walls) [92] may be considered as an endogenous gut prebiotic product that can stimulate the innate immune system against pathogenic microorganisms [93]. In gut bacteria that inhabit the intestines of humans, when considering the bacterial cell wall component peptidoglycan, it is viewed as a key bacterial cell component. In immunology, peptidoglycans are conserved pathogen-associated molecular patterns (PAMP) for which the innate immune system has evolved sensing mechanisms [93]. Further, peptidoglycans have been reported to be a direct target for innate immune responses and for regulating the accessibility of other PAMP for additional innate immune responses.

## 3. Functional Foods for T2DM

From birth there is a complex and dynamic microbial assault on all mucosal surfaces of the host to achieve colonization with an obligatory progression to immunological tolerance and metabolic homeostasis [94]. These events can be enhanced with the administration of functional foods such as prebiotics and probiotics that have been reported to induce immunomodulatory and metabolic balancing activities in the intestines [86,95]. Different types of prebiotics can influence the intestinal microbiome by stimulating the growth of various indigenous bacterial communities. The combined research evidence from laboratory, animal, and human studies has shown that when prebiotics have been administered, there has been a subsequent gut enrichment of commensal species that promote health, such as those from the genera *Lactobacillus* and *Bifidobacterium* [96,97].

Oligosaccharides have been reported to improve inflammatory responses in the gut [98]. The intestinal microbiome has been reported to be significantly associated with gross adverse shifts in the structure of the intestinal microbiome. Furthermore, reports showed that the genera *Ruminococcus*, *Fusobacterium*, and *Blautia* were positively associated with T2DM, whereas the genera *Bifidobacterium*, *Bacteroides*, *Faecalibacterium*, *Akkermansia,* and *Roseburia* were negatively associated with T2DM [25]. Studies have shown that oligosaccharides in the intestine can be fermented and utilized by various bacteria, including *Streptococcus* [99], *Escherichia coli* [100], and *Clostridium* species [101]. A small clinical study explored the adjunctive administration of oligofructose with sulfonylurea [28], which increased the abundance of *Bifidobacterium*, a genus of intestinal bacteria, and reported the improvement of glycemic control in T2DM [102]. *Bifidobacterium* are reported to be acetate and propionate producers of SCFAs [103]. As such, intestinal bacteria from the genera *Faecalibacterium*, *Akkermansia*, and *Roseburia* utilize these SCFAs as energy sources while secreting butyrate, subsequently improving insulin resistance and glucose metabolism.

The adjunctive treatment for 6 months with oligofructose prebiotic for patients diagnosed with T2DM while prescribed sulfonylurea monotherapy resulted in clinically significant improvements in fasting blood glucose, postprandial blood glucose, area under the curve for serum glucose, and HbA1c. These improvements coincided with an increase in the endogenous insulin and c-peptide levels, collectively leading to significant improvements in pancreatic beta cell function. These improvements in glycemic control were associated with changes in the gut microbiome dynamics and an increase in beneficial and a reduction in detrimental bacteria that have previously been found to be associated with T2DM (Table 2) [28].

**Table 2 biomolecules-13-01307-t002:** Adjuvant prebiotics with and without pharmacotherapy for the management of T2DM. human studies.

PREBIOTICS (n) REFERENCE	TREATMENT TIME	DOSE ADMINISTERED	OUTCOMES
**STUDIES with Prebiotics and T2DM**			
FOS * [n = 27 W] [47]	8-weeks	10 g/day O.D.	↓ Fasting plasma glucose ↓ Glycosylated hemoglobin ↓ Interleukin-6 ↓ Tumor Necrosis Factor α ↓ Plasma Lipopolysaccharide
RS [n = 17] [104]	12-weeks	40 g/day O.D.	↓ Postprandial Glucose (meal tolerance test) ↑ Glucagon-Like Peptide-1 ↓ Tumor Necrosis Factor
AX [n = 15] [105]	5-weeks	49.2 g/day	↓ Fasting Serum Glucose Level. ↓ Serum Glucose ↓ Insulin Level (2 h after oral glucose intake)
**STUDIES with Prebiotics + Pharmacotherapy and T2DM**			
Prebiotic Fiber + Sulfonylurea [n= 30] [28]	24-weeks	Sulfonylurea + OZ101 an OLF 13.5 or 27 g/d T.I.D.	Low dose of 13.5 g/T.I.D. associated… − improved beta-cell function stabilization of glycemic control over 24 weeks.
Prebiotic Fiber + Metformin [n = 9 with 6 eligible] [106]	1-week cross-over 4-week follow-up	Metformin (850 mg) O.D. increased B.I.D. + Prebiotic Fiber (BiomeBliss) O.D.	Modest shifts in microbial composition − Proof-of-Concept feasibility for adjunctive metformin therapy

* O.D. = once per day; B.I.D. = twice per day; T.I.D. = three times per day; FOS = fructo-oligosaccharides; RS = resistant Starch; AX = arabinoxylan; OLF = oligofructose; ↑ = increased; ↓ = decreased.

## 4. Modulation of the Intestinal Microbiome in T2DM

The disruption of the epigenome poise may be causal for several pathologies and is a contributing factor to obesity and T2DM [107]. It is known that the epigenome includes DNA methylation, histone modifications, and RNA-mediated processes, where these mechanisms control gene activity and physiological development [108]. This can be exemplified through the adipose tissue, skeletal muscle, pancreatic islets, liver, and blood relations to obesity and T2DM [108,109].

In a sequence of important studies with T2DM patients and intestinal microbiota, researchers have reported the characterization of the intestinal microbiota with this metabolic disease (Table 3). Moderate gut microbial dysbiosis was applied to characterize the gut microbiota. A functional analysis has shown that glucose membrane transport, methane metabolism, heterogeneous biomass degradation, branched-chain amino acid transport and metabolism, and sulphate reduction pathways were enriched in patients with T2DM.

**Table 3 biomolecules-13-01307-t003:** The complex and progressive characterization of the intestinal microbiome associated with T2DM.

Patients with T2DM * (n) [Reference]	Faecal Microbiome Analysis	Gut Microbiome Characterizations with T2DM
(n = 18/36 M|T2DM) [110]	real-time quantitative PCR (qPCR)	Plasma glucose association with ↓ abundance *Firmicutes* phylum ↓ abundance *Clostridia* class ↑ abundance *Betaproteobacteria* class
	subgroup (n = 20) tag-encoded amplicon pyrosequencing of the V4 region of the 16S rRNA gene	+ve association with ratios of *Bacteroidetes* > *Firmicutes* *Bacteroides-Prevotella* > *C. coccoides-E. rectale*
(n = 345 and 23 T2DM) [111]	MGWAS	↓ abundance of butyrate-producing bacteria *Clostridiales* sp. SS3/4|*F. prausnitzii*| *Roseburia intestinalis*|*Eubacteriumrectale*| *Roseburia inulinivorans* T2DM characterized by a moderate degree of intestinal microbial dysbiosis
		↑ abundance opportunistic pathogen bacteria *Bacteroides caccae*|*Clostridium hathewayi*| *Clostridium symbiosum*|*Eggerthella lenta*| * Clostridium ramosum*|*Escherichia coli* ↑ mucin-degrading bacteria *Akkermansia muciniphila* ↑ sulfate-reducing bacteria *Desulfovibrio* sp. 3_1_syn3
(n = 2595 W| 9.5% T2DM 14.4% IGT [112]	SgSeq faecal meta-genome sample n = 145 W normal|impaired| diabetic glucose control	Further confirmation − functional alterations of the gut microbiome reflecting changes in the intestinal environment of T2DM Patients with T2DM ↑ abundance 4 *Lactobacillus* spp. *L. gasseri* highest ↓ abundance 5 *Clostridium* spp. *Roseburia*| * Faecalibacterium prausnitzii*| − highly discriminant for T2DM − butyrate producers − linked with improved insulin sensitivity

* T2DM = type 2 diabetes mellitus; SgSeq = shotgun sequencing; MGWAS = metagenome-wide association study; M = men; W = women; ↑ = increased; ↓ = decreased.

T2DM and obesity are strongly linked with adiposopathy, along with adipose tissue dysfunction, which has been reported as a major factor in the etiology of metabolic dysfunction conditions [113]. Metabolic dysfunction promotes dysregulated glucose homeostasis and impaired adipogenesis, leading to the accumulation of ectopic fat, insulin resistance, and chronic inflammation [113]. The intestinal inflammatory cascade that is progressed with T2DM disrupts intestinal barrier homeostasis, increasing gut permeability, with a consequent increased risk of endotoxemia with the translocation of endotoxins, such as lipopolysaccharides (i.e., LPS), from Gram-negative bacteria (e.g., *Clostridium perfringens*) [114]. This significantly affects glucose metabolism and insulin resistance [114].

A recent systematic review analysed studies that had assessed the serum concentrations of the endotoxin lipopolysaccharide (LPS) or that of the lipopolysaccharide-binding protein (LBP) in diabetic patients and compared them to the serum of healthy individuals [115]. The review presented significant variability in the estimates that the studies reported of metabolic endotoxemia [115]. Most studies reported higher LPS or LBP concentrations in diabetic patients than in those of the healthy controls. Patients with type 1 diabetes mellitus (T1DM) and T2DM presented with higher mean fasting levels of LPS of 235.7% and 66.4%, respectively, as compared with non-diabetic patients. Advanced complications (e.g., such as macroalbuminuria) and disease onset were reported to exacerbate endotoxemia. Furthermore, antidiabetic medications interestingly decreased fasting LPS concentrations [115]. Among the medications, rosiglitazone and insulin presented with higher and lower effects, respectively, when compared with other pharmaceutical treatments. T1DM and T2DM appeared to increase metabolic endotoxemia [115]. The authors concluded that numerous confounders (i.e., diet, age, medication, smoking, and obesity) can influence both diabetes and endotoxemia expression [115].

An early review [116] reported that when LPS is present in the intestinal lumen, it can translocate the intestinal mucosal barrier and subsequently into the systemic circulation, causing metabolic endotoxemia [116]. LPS has the capacity to bind to toll-like receptor 4 (i.e., a receptor that belongs to the family of pattern recognition receptors (PRRs)).

There is subsequent activation of pro-inflammatory activity that can alter several stages of insulin signaling. Furthermore, the review reports that chronic exposure to the LPS endotoxin may contribute to weight gain and the progression of T2DM [117].

Obese and diabetic individuals have an increased risk of presenting with plasmatic LPS levels [116]. The increase in the number of Gram-negative bacteria in the gut microbiota, the reduction in gut mucosal integrity, and the consumption of high-fat diets increase plasmatic lipopolysaccharide levels [116]. Hence, it is therefore plausible to conclude that the type of diet that patients consume may modulate the composition of the intestinal microbiome that could ultimately improve intestinal barrier mucosal integrity and decrease the occurrence of endotoxemia and its postprandial inflammatory effects. This combined activity leads to adequate insulin signaling [116]. The authors note that there are few studies that have evaluated the influence of nutrients and/or specific food types on metabolic endotoxemia [116].

In a recent study linked to endotoxemia, Cox and colleagues [118] investigated the relationships that can exist between the intestinal microbiome, intestinal permeability (i.e., the intestinal epithelial barrier disruption), and inflammatory responses as multi-factorial risk factors for obesity-associated metabolic diseases. The study aimed to investigate the associations between intestinal epithelial cell permeability and T2DM [118].

The primary outcome data demonstrated that there were differences in the measures of intestinal epithelial cell barrier permeability between individuals with and without T2DM [118]. The authors suggested that using intestinal epithelial cell barrier permeability measures as a tool for predicting T2DM risk has biological plausibility. Consequently, a recent study used plasma zonulin levels (zonulin is a protein modulator of intercellular tight junctions in the intestines) as a non-invasive biomarker of intestinal permeability in women diagnosed with gestational diabetes [119]. Although further studies are warranted, this initial study showed that there was a positive correlation between plasma zonulin levels and body mass index, creatinine, fasting plasma glucose, baseline, first hour, and two-hour glucose levels and the OGTT, haemoglobin A1C (HbA1_C_), homeostatic model assessment for insulin resistance (HOMA-IR), and alanine aminotransferase (ALT) levels. The authors suggested that the findings could promote zonulin as a non-invasive biomarker involved in the pathogenesis of gestational diabetes [119].

Butyrate, a 4-carbon SCFA produced by intestinal bacteria [120] via glycolysis during the intestinal microbial fermentation of an undigested carbohydrate, has been shown to be a key facilitator in the assembly of the intestinal epithelial barrier [121]. Butyrate exhibits numerous biological activities in different experimental settings, including energy homeostasis, glucose and lipid metabolism, inflammation, redox and neural signaling, and epigenetic modulation [122]. Importantly, butyrate has been reported to down-regulate inflammation by inhibiting the growth of pathobionts, increasing mucosal barrier integrity, encouraging obligate anaerobic bacterial dominance, and decreasing oxygen availability in the gut [122].

Furthermore, butyrate can decrease excessive inflammation through the modulation of immune cells, e.g., by increasing the functionalities of M2 macrophages and regulatory T cells and inhibiting infiltration by neutrophils [122]. Therefore, bacterial species known to be butyrate-producers or dietary components that will enhance the molar ratio of butyrate production are considered as important in maintaining an effective gut barrier and improving glucose metabolism and insulin resistance [123].

## 5. Discussion

In this narrative review, we have focused on the adjuvant role that functional foods such as prebiotics may provide to the intestinal microbiome to improve metabolic dispositions in T2DM. In addition, while the dietary approach to modulating the intestinal microbiome with the administration of probiotics has a long history, there has always been probiotic survivability and viability issues with the administration of various probiotic formulations that have been difficult to ascertain.

We have previously investigated, through clinical studies, that functional foods with prebiotics and probiotics, co-administered with standard medications, were effective for the management of T2DM. These studies confirmed that the co-administration of an oligofructose prebiotic with sulfonylurea [28] and a multi-strain probiotic plus metformin [27] improved glycemic control in patients diagnosed with T2DM.

The intestinal microbiome has been suggested to have a significant impact on transforming T2DM [124]. The review concluded that there was biologically plausible evidence for the gut microbiome’s ability to influence and improve T2DM symptomatology. Moreover, the authors of the review partially demonstrated the benefit that probiotics, prebiotics, and synbiotics may afford to patients diagnosed with pre-diabetes. The conclusion was that any beneficial effects were centred on modulating the abundance of the intestinal microbiome [124].

Animal and human studies investigating *Clostridium* species that mostly utilize indigestible polysaccharides by principally fermenting carbohydrates, proteins, and organic acids have shown that these bacteria produce SCFAs of acetic acid, propionic acid, and butyric acid, thereby affording numerous benefits to gut health through intestinal actions which can translate to enhanced management of T2DM [125]. Consequently, research continues to report the importance of SCFAs in the gut, such as that attributed to butyrate, a molecule with ubiquitous epigenetic effects on the gut [120,126]. Among the gram-positive anaerobic bacteria, butyrate-producing bacteria are widely distributed in the intestines. Two of the most important groups of intestinal microbes are *Faecalibacterium prausnitzii* in the *Clostridium leptum* cluster (i.e., *Clostridial* cluster IV) and *Eubacterium rectale/Roseburia* spp. in the *Clostridium coccoides* (i.e., *Clostridial* cluster XIVa) cluster of *Firmicutes* [127]. These groups of intestinal bacteria typically account for approximately 5–10% of the total bacteria that have been detected in the stool samples of healthy adult humans [127]. These groups of intestinal butyrate-producing bacteria are widely distributed across several groups that include the clusters IX, XV, XVI, and XVII [127].The importance of the presence of *Bacteroides* resides in their capacity to ferment carbohydrates. Members of the taxa *Bacteroides* and other intestinal bacteria produce a significant mixture of fatty acids, which can be utilized by the host as an energy source [128], and this may have favourable sequelae in T2DM. An early review reported that the dominant intestinal bacteria consist of anaerobes from the taxa *Bacteroides*, *Bifidobacterium*, *Eubacterium*, *Lactobacillus, and Streptococci* [129]. However, *Enterobacteria* were cited as being usually found in a lesser abundance [129]. A recent report investigated the universal healthy intestinal microbiome from numerous geographical areas, establishing that one of the abundant clusters, namely *Bacteroides*, results in dominant bacterial signatures [130].

Furthermore, a cluster of bacteria consisting of *Ruminococcus* and *Blautia* were also reported, and the significance of this is that these clusters have significant positive associations with T2DM [25]. *Blautia* is widely distributed in mammalian faeces and the intestines. As a dominant genus in the intestinal microbiota, there is a significant correlation between its host physiological and metabolic dysfunctions with obesity, diabetes, and various inflammatory and metabolic diseases due to its antibacterial activity against specific commensal microorganisms [131]. In addition, members of the taxa *Ruminococcus* also have a relevant importance to T2DM, as the members of this taxon breakdown cellulose that produces methane with the accumulation of a reserve of glucose polymers important in glucose homeostasis [25,130]. Reports that show that galactooligosaccharide reduced the abundance of the *Ruminococcus* taxa in the intestines may constitute a significant clinical step in prescribing specific prebiotic fibers, beneficially improving the glycemic control of T2DM [132] by reducing the abundance of intestinal microbes that promote diabetes. The associated increase in *Bacteroidaceae* levels in sulfonylurea monotherapy patients taking adjuvant oligofructose prebiotics provides further support to the posit of a beneficial role of this family of bacteria in T2DM [28].

Patients without metabolic diseases, in contrast to studies that have described the effects that prebiotic fibers may have on the composition of the intestinal microbiota, show consistently reported increases in the abundance of *Bifidobacterium* [133,134,135] and/or *Lactobacillus* species [135,136]. Intestinal microbial cross-feeding mechanisms in the colon are thought to form the basis of butyrate production [137], a recognised functional characteristic of several colon bacteria that, in part, explains *Bifidobacterium* competitiveness and butyrate production [137]. In addition, the specificity of polysaccharide use by the colon microbiota may determine diet-induced alterations in the microbiota and consequent advantageous metabolic effects in patients with T2DM. Certainly, supplementation with nondigestible polysaccharides of plant origin is important for the enrichment of the intestinal microbiota with *Lactobacillus* and *Bifidobacterium* species, which ferment these compounds into SCFAs [103,138]. Of further importance are the genera *Bifidobacterium*, *Bacteroides*, *Faecalibacterium*, *Akkermansia,* and *Roseburia* that have been reported to be significantly and negatively associated with T2DM [25]. Specifically, the genus *Faecalibacterium* consists of abundant butyric acid-producing bacteria that colonize the human intestines and display an overall anti-inflammatory effect with the potential of re-regulating adverse inflammatory responses in the gut [139], a prevalent adverse factor that is present in patients diagnosed with T2DM. Similarly, *Roseburia* also shows stable and relatively abundant levels in the gut that show favourable sequelae in glucose metabolism in T2DM. Members of the *Roseburia* genus likely play a major role in maintaining gut health and immune defence by maintaining regulatory T-cell homeostasis, predominantly through the production of butyrate [140]. Conversely, a recent study investigating butyrate-producing gut bacteria and insulin homeostasis showed that not all butyrate-producing intestinal bacteria can benefit patients diagnosed with T2DM [141]. The authors reported that although most butyrate producers analysed appear to be metabolically beneficial for T2DM, this was not the case for all such gut bacteria. In contrast, *Flavonifractor* (i.e., Gram-positive bacteria belonging to the *Clostridiales* order) was associated with lower insulin sensitivity and disposition index and a higher prevalence of dysglycemia [141]. Hence, a biologically plausible posit ensues that clinical measures to treat or prevent T2DM should be targeted to specific butyrate-producing taxa rather than all gut microbiome butyrate producers [141].

## 6. Conclusions

The clinical use of supplemented fibers, namely oligofructose, with hypoglycemic medications such as sulfonylureas could improve health outcomes in patients diagnosed with pre-T2DM and early-stage T2DM. The intestinal benefits have been attributed to an increased abundance of butyrate-producing intestinal microbes, including *F. prauznitzii*, that improve intestinal health maintenance (e.g., reducing intestinal epithelial barrier permeability and pro-inflammatory actions in the gut) and immune defence by maintaining regulatory T-cell homeostasis. Moreover, this bacterium has been promoted as a new generation probiotic [142]; therefore, a prebiotic fiber, such as oligofructose, that can enhance the abundance of such beneficial bacteria in the gut could be posited to improve T2DM metabolic dispositions when subsequently combined with sulfonylurea. Furthermore, it is noteworthy that well-balanced glucose homeostasis may require the presence of several specific bacterial species that work together to provide a number of functions, each of which with distinct properties.

## Figures and Tables

**Figure 1 biomolecules-13-01307-f001:**
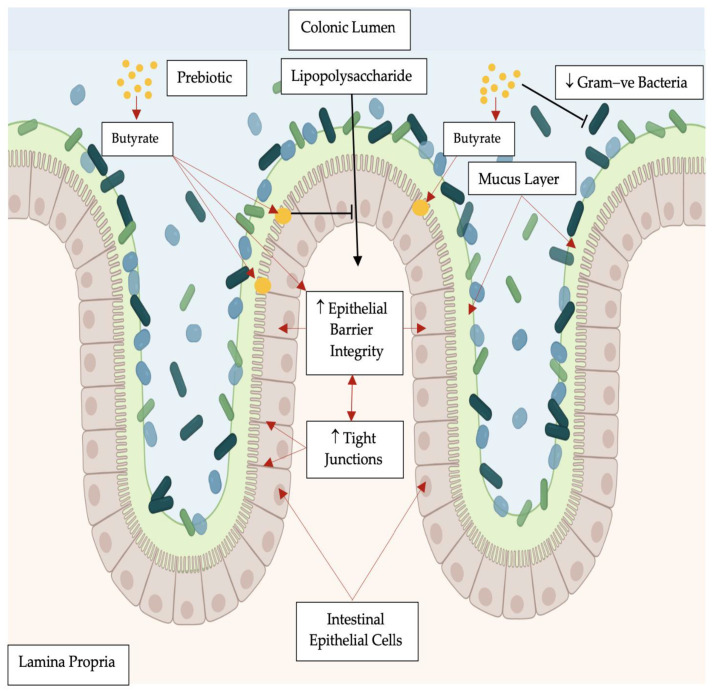
Prebiotics increase the concentration of luminal butyrate [85], facilitate tight junction barrier assembly, improve the integrity of the intestinal epithelium that inhibits the translocation of endotoxins (e.g., lipopolysaccharide) and decrease the abundance of Gram-negative bacteria [84]; a large proportion of the of the indigenous bacteria are Gram-negative, which accounts for the high LPS load that the colonic gut can experience [86].

## Data Availability

Not applicable.

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
