# Peer review of "Prebiotics Progress Shifts in the Intestinal Microbiome That Benefits Patients with Type 2 Diabetes Mellitus"

_biomolecules, 2023, doi:10.3390/biom13091307_

Round 1
Reviewer 1 Report
Luis Vitetta and co-authors have reviewed on the topic “Prebiotics Progress Shifts in the Intestinal Microbiome that Benefits Patients with Type 2 Diabetes Mellitus”.
This review has discussed numerous studies conducted in different areas related to the topic. However, a more organized writing will add clarity for the readers who are new to the topic.
The main theme of the article is that prebiotics benefits diabetes mellitus patients by shifting their microbiome profile from dysbiosis.
Under a topic like this, it is more appropriate to start describing the pathogenesis of T2DM mediated through microbiome dysbiosis before describing how prebiotics influence T2DM.
However, in this manuscript, characterization of intestinal microbiome with T2DM is given in Table 3 whereas Table 2 has already mentioned how adjuvant prebiotics with and without pharmacotherapy helps T2DM management.
I would recommend to re-write the review with sub-sections arranged in this following order.
1. Pathogenesis of Diabetes mellitus: link between gut microbiome dysbiosis and insulin resistance.
This is described in the current manuscript from line 210-222.
Modify lines 215-218: both sentences essentially say the same thing.
Important points missing under this section:
How is endotoxemia linked to the molecular mechanism of insulin resistance?
Consider the following reference instead of reference No:75 to explain this.
- The influence of endotoxemia on the molecular mechanisms of insulin resistance PMID: 22732959
Table: The complex and progressive characterization of the intestinal microbiome associated with T2DM.
Recommendation: Include a schematic diagram showing the molecular mechanism linking gut microbiome dysbiosis and insulin resistance in T2DM.
Further references suggested regarding the loss of intestinal permeability in DM
Circulating zonulin levels in newly diagnosed Chinese type 2 diabetes patients.
Diabetes Res Clin Pract 2014;106(2):312–8.
Demir E, et al. Plasma zonulin levels as a non-invasive biomarker of intestinal permeability in women with gestational diabetes mellitus. Biomolecules 2019;9(1):24.
Increased intestinal permeability as a risk factor for type 2 diabetes. PMID: 27745826
2. Prebiotics and intestinal bacteria. (Line 50-109)
Comment: This review is intended to be about the health benefits of prebiotics. However, there is no clear distinction explained between dietary fiber and prebiotics.
One suggested reference for introducing the term prebiotic and further explanation:
Fiber and Prebiotics: Mechanisms and Health Benefits. PMID: 23609775
Figure: Prebiotics increase the concentration of luminal butyrate, improves the integrity of the intestinal epithelium inhibiting the translocation of endotoxins (e.g., lipopolysaccharide) and decreases the abundance of Gram-negative bacteria.
Question: How is the abundance of Gram-negative bacteria decreased by prebiotics?
Recommendation:
Modify the figure by including more details about the role of SCFA (butyrate) in reducing the intestinal permeability through acting at Tight junctions.
Peng L, et al. Butyrate enhances the intestinal barrier by facilitating tight junction assembly via
activation of AMP-activated protein kinase in Caco-2 cell monolayers. J Nutr 2009;139(9): 1619–25.
Table: Prebiotics Shift the Intestinal Microbiome: Human Studies
Make sure that all short forms of prebiotics are explained after the table.
Expand abbreviated form of AX, GOS.
Correct the spelling of Corn starch in the last row. Also, last row is missing information in 2nd, 3rd and 4th column. In the first column it is given digestible corn starch and potato. In the last column it is given ‘tapioca’. Is this correct?
In the Reference 33 data, it is written “Shift of B:F ratio”. Comment on firmicutes/Bacteroides ratio. Is there a recommended healthy ratio?
Question: The review mentions multiple times that butyrate is beneficial for maintaining the integrity of intestinal epithelium. Most of the studies reviewed shows increase in Bifidobacterium and lactobacilli. At the same time, it is also written that members of these genera do not form butyrate.
What bacteria are exactly forming butyrate?
Does prebiotics shift the microbiome to butyrate-producers?
Line 51-52. Check grammar.
Line 100. “act as a histone”
Line 105-108- Check grammar
Line 109- 126: Is this relevant to mention under this topic?
3. Prebiotics for T2DM (Actual topic)
Line 130- 139: Repetition of the ideas given under “Prebiotics and intestinal bacteria”.
Line 140-152: This should come under the section “Pathogenesis of Diabetes mellitus: link between gut microbiome dysbiosis and insulin resistance”. Several ideas are repeated.
Line 153- 168: Important part under the topic.
Line 169-174: Direct effect of Metformin on intestinal microbiome.
Line 176- 183: Important part that must be elaborated in this review.
Table: Adjuvant prebiotics with and without pharmacotherapy for the management of T2DM: Human studies.
Please write more clearly what are the outcomes of studies with Prebiotics and Pharmacotherapy.
Line 224- 234: This part comes under this section.
Line 249-262: Comes under Prebiotics for T2DM.
Line 256-258: A small clinical study published the adjunctive administration of an oligofructose with sulfonylurea [7] increased the abundance of Bifidobacterium a genus of intestinal bacteria, reporting the improvement of glycemic control in T2DM [87].
This sentence is about a study from the authors of this review.
Still a wrong reference is given:
87: “Targeting gut microbiota: Lactobacillus alleviated type 2 diabetes via inhibiting LPS secretion and activating GPR43 pathway. Journal of Functional Foods. 2017; 561-570.”
Many points are repeated in the discussion and conclusion parts. It is recommended to rewrite these sections in a more comprehensible manner.
Question:
Are there studies that shows that prebiotic intake by diabetes patients with or without pharmacotherapy resulted in the shift of their microbiome from dysbiosis and this resulted in increase in SCFAs like butyrate which finally reduced the symptoms of T2DM.
Moderate editing of English grammar is recommended.
Author Response
We thak the reviewer for his/her comments and queries.
Please see attached document for the rebuttals to reviewer 1 queries.
Thank you
Luis Vitetta on behalf of all authors.

Reviewer 2 Report
In this review, Vitetta and colleagues discuss prebiotics progress shifts in the intestinal microbiome that benefits patients with type 2 diabetes mellitus. The manuscript is well-written and well-organized. However, I have only one minor point that could improve this manuscript.
- In this context, the Authors might want to add more studies showing intestinal glucotoxicity. It has been reported indeed that glucotoxicity impairs L-cell differentiation, which could be associated with decreased intestinal stem cell proliferative capacity (PMID: 34206340). It would be interesting to understand whether treatments with prebiotics might reduce intestinal glucotoxicity and thus ameliorate the quality of life of patients suffering from type 2 diabetes mellitus.
Minor editing
Author Response
We thank reviewer 2 for his / her comments.
Please see attached document for the rebuttal to the query from this reviewer.
Thank you
Luis Vitetta on behalf of all authors.
